# Stereoselective alkoxycarbonylation of unactivated C($sp^3$)–H bonds with alkyl chloroformates via Pd(II)/Pd(IV) catalysis

Gang Liao[1], Xue-Song Yin[1], Kai Chen[2], Qi Zhang[1], Shuo-Qing Zhang[1] & Bing-Feng Shi[1]

Several examples on Pd-catalysed carbonylation of methyl C($sp^3$)–H bonds with gaseous CO via Pd(II)/Pd(0) catalysis have been reported. However, methylene C($sp^3$)–H carbonylation remains a great challenge, largely due to the lack of reactivity of C–H bonds and the difficulty in CO migratory insertion. Herein, we report the stereoselective alkoxycarbonylation of both methyl and methylene C($sp^3$)–H bonds with alkyl chloroformates through a Pd(II)/Pd(IV) catalytic cycle. A broad range of aliphatic carboxamides and alkyl chloroformates are compatible with this protocol. In addition, this process is scalable and the directing group could be easily removed under mild conditions with complete retention of configuration.

[1] Department of Chemistry, Zhejiang University, Hangzhou 310027, China. [2] Division of Chemistry and Chemical Engineering, California Institute of Technology, Pasadena, California 91125, USA. Correspondence and requests for materials should be addressed to B.-F.S. (email: bfshi@zju.edu.cn).

Over the past few decades, Pd-catalysed C–H functionalization has emerged as a powerful tool for the direct conversion of ubiquitous C–H bonds into diverse functional groups[1–16]. Among various C–H functionalization reactions, the direct alkoxycarbonylation of C–H bonds is particularly valuable[17–21], since the resulting products, esters, are among the most important functional groups that appear commonly in agrochemicals, fine chemicals, natural products and pharmaceuticals.

Although Pd-catalysed carbonylation of aromatic C–H bonds has been extensively investigated, the direct carbonylation of aliphatic C–H bonds is limited and still represents a tremendous challenge in organic synthesis[17–21]. Pd-catalysed carbonylation of methyl C($sp^3$)–H bonds of aliphatic amides or amines with CO for the synthesis of succinimides or lactams has been achieved recently[22–27]. Generally, these reactions proceed through a Pd(II)/Pd(0) catalytic cycle. While these elegant methods are efficient to introduce carbonyl groups and have greatly enriched the reaction scope, the use of CO is still relatively inconvenient on laboratory-scale due to its gaseous form, toxic nature and flammability. In addition, the carbonylation reactions were limited to those functionalizing methyl C($sp^3$)–H bonds (Fig. 1a)[22–27]. The analogous carbonylation of methylene C($sp^3$)–H bonds, which are more inert and sterically hindered than methyl C($sp^3$)–H bonds, remains unexplored[28,29].

Pd-catalysed alkoxycarboxylation of C($sp^2$)–H bonds with other carbonyl reagents, such as potassium oxalate monoester[30], DMF[31], formates[32], azodicarboxylates[33,34], α-keto esters[35], glyoxylates[36] and oxaziridine[37] has been reported. Ru-catalysed alkoxycarbonylation of 2-arylpyridines with alkyl chloroformates was disclosed by Kakiuchi and co-workers[38]. Inspired by these excellent precedents and based on our recent work on Pd-catalysed C($sp^3$)–H functionalization[39–41], we were eager to develop the catalytic carbonylation of methylene C($sp^3$)–H bonds with less toxic, more easy to handle and readily available carbonyl reagents.

Herein, we report the stereoselective and site-selective alkoxycarbonylation of unactivated C($sp^3$)–H bonds with alkyl chloroformates through a Pd(II)/Pd(IV) catalytic cycle (Fig. 1b). This reaction is environmentally friendly and operationally simple. A broad range of aliphatic carboxamides and alkyl chloroformates are compatible with this protocol. In addition, this process is scalable and the directing group could be easily removed under mild conditions with complete retention of configuration, thus providing a convenient strategy for the stereoselective synthesis of orthogonally protected aspartic acid derivatives[42–45]. Compared with the well-established C–H carbonylation with CO via Pd(II)/Pd(0) catalysis, the direct alkoxycarbonylation of unactivated C($sp^3$)–H bonds through a Pd(II)/Pd(IV) catalytic cycle provides a new mode of reaction and might offer a distinct platform for reaction development.

## Results

**Proof of concept on methylene C($sp^3$)–H alkoxycarbonylation.** We began our investigation by using N-phthaloyl phenylalanine derivative **1a** bearing an 8-aminoquinoline (AQ) auxiliary as a model substrate. This auxiliary was first introduced by Daugulis and has been proved to be effective in the direct functionalization of methylene C($sp^3$)–H bonds[46–50]. We first explored the carbonylation with carbon monoxide through the traditional Pd(II)/Pd(0) pathway. Previously, we have found that the reaction of N-phthaloyl phenylalanine derivative **1a** and stoichiometric Pd(OAc)₂ could form the stable palladacycle **I** in MeCN (Fig. 2a)[51,52]. However, when we treated palladacycle **I** with carbon monoxide under various conditions, no desired carbonylation product 3ab was observed. Complex **II** with CO coordinated as an L-type ligand was obtained as a pale yellow solid in 80% yield (Fig. 2b). This complex showed unexpected resistance to migratory insertion under various conditions, and proved stable to air and moisture, withstanding shelf storage without noticeable decomposition. The coordination of CO as a neutral ligand without migratory insertion was confirmed by the characteristic infrared absorption of terminally coordinated CO ligand (2095 cm⁻¹, Supplementary Fig. 2), and, the ease formation of complex **III** via ligand exchange with pyridine,

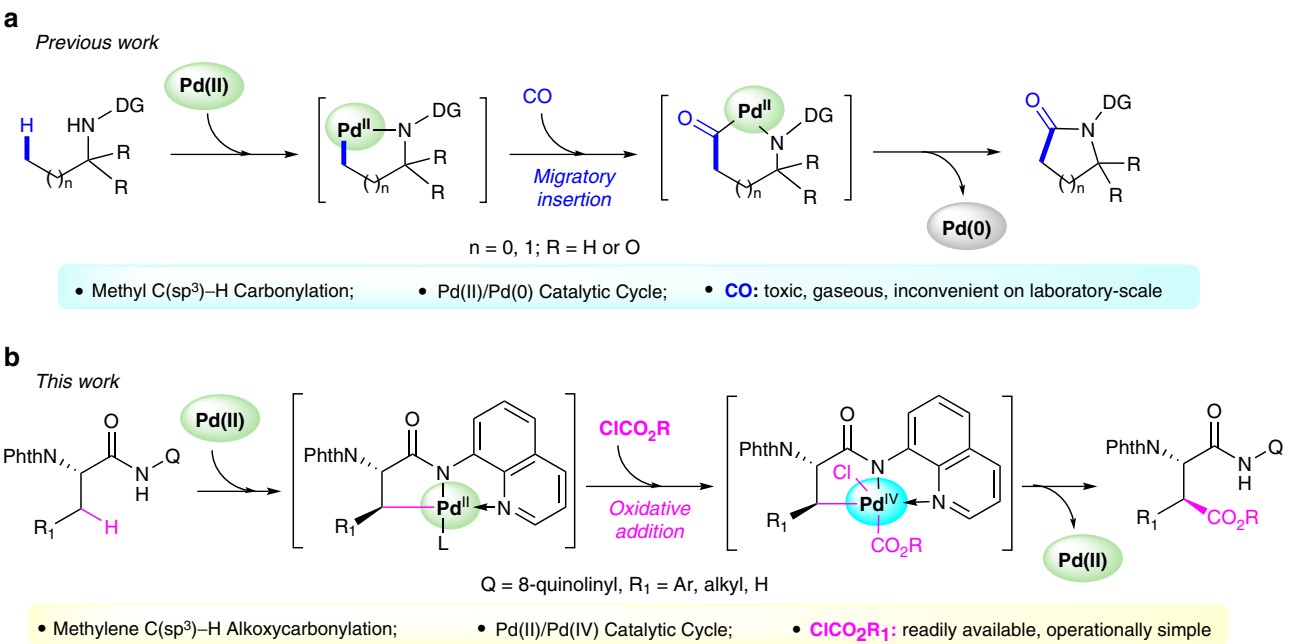

**a** Previous work

**b** This work

Q = 8-quinolinyl, R₁ = Ar, alkyl, H

**Figure 1 | Pd-catalyzed carbonylation of C($sp^3$)–H bonds.** (**a**) Previous reports on Pd-catalyzed carbonylation of methyl C($sp^3$)–H bonds with carbon monoxide through a Pd(0)/Pd(II) catalytic cycle. (**b**) Our work on Pd-catalyzed alkoxycarbonylation of unactivated C($sp^3$)–H bonds with alkyl chloroformates through a Pd(II)/Pd(IV) catalytic cycle.

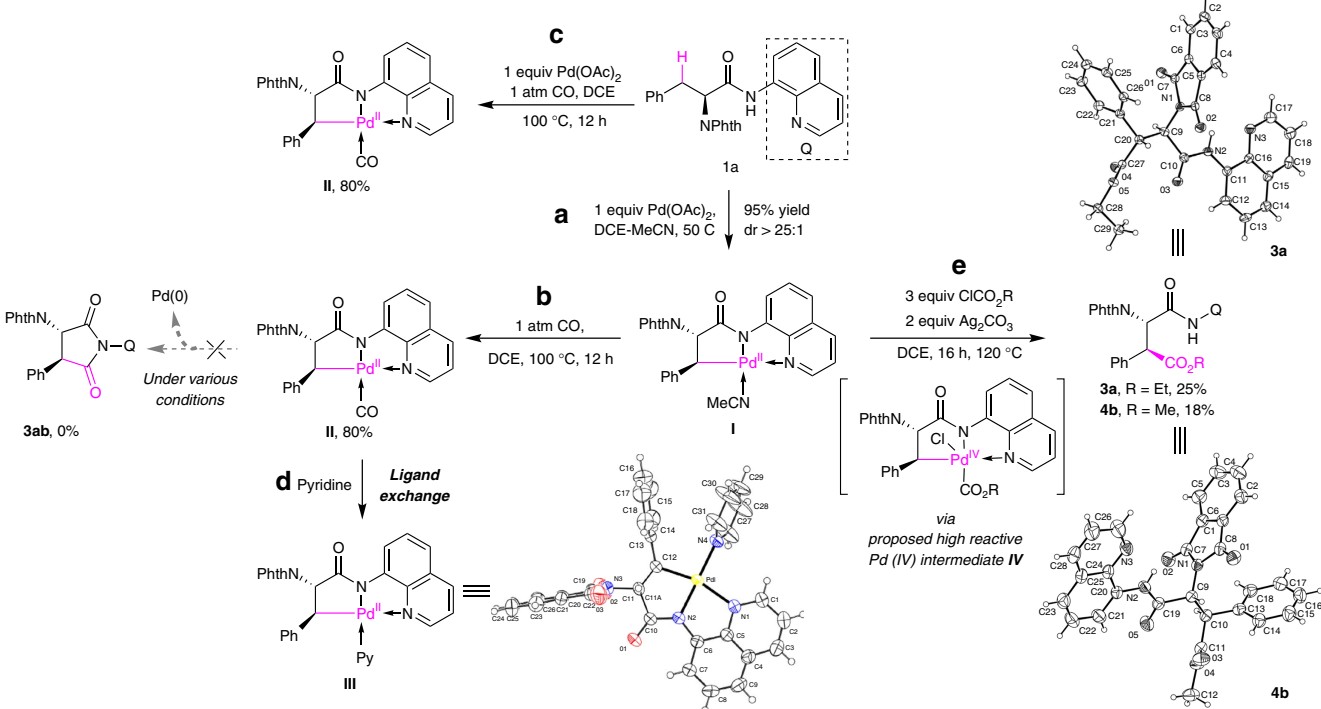

**Figure 2 | Proof of concept on alkoxycarbonylation of methylene C($sp^3$)–H bonds via Pd(II)/Pd(IV) catalysis.** (**a**) Synthesis of palladacycle **I**. (**b**) Stoichiometric reaction of palladacycle **I** with carbon monoxide. (**c**) Synthesis of CO-coordinated Pd(II) complex **II**. (**d**) Structure confirmation of complex **II** via the transformation to complex **III**. The structure of complex **III** was unambiguously confirmed by single-crystal X-ray diffraction. (**e**) Stoichiometric reaction of palladacycle **I** with ClCO$_2$R. The structure of compounds **3a** and **4b** was confirmed by single-crystal X-ray diffraction.

which was characterized by X-ray crystallography (Fig. 2d). It is noteworthy that complex **III** could also be generated via the reaction of **1a** with 1 equivalent of Pd(OAc)$_2$ in a mixture of DCE and pyridine (see Supplementary Methods for details). Moreover, the reaction of **1a** with stoichiometric Pd(OAc)$_2$ under 1 atm CO could also give complex **II** in 80% yield without the detection of any carbonylated product **3ab** (Fig. 2c). Thus, the carbonylation of β-methylene C($sp^3$)–H bonds of **1a** through Pd(II)/Pd(0) was unfeasible due to the difficulty in CO migratory insertion and subsequent reductive elimination[53,54].

It has been proven that high-valent Pd(IV) species undergo facile reductive elimination[55–59]. Therefore, we speculated that the use of alkyl chloroformates as carbonylation reagent might enable the desired alkoxycarbonylation via oxidative addition of palladacycle **I** to form a highly reactive Pd(IV) intermediate **IV**, which could then undergo reductive elimination to give the corresponding ester (Scheme 1, Pd(II)/Pd(IV) pathway). To our delight, treatment of complex **I** with 3 equiv. of ClCO$_2$Et (**2ab**) or ClCO$_2$Me (**2b**) in the presence of 2 equiv. of silver carbonate gives the expected alkoxycarbonylation product **3a** and **4b** in 25% and 18% yield, respectively. The relative and absolute stereochemistry of **3a** and **4b** was unambiguously determined by X-ray crystallography (Fig. 2e). Inspired by this promising result, we next sought to identify suitable reaction conditions to render this reaction catalytically (Table 1). When **1a** was treated with 10 mol% Pd(OAc)$_2$, 3 equiv. of ClCO$_2$Et and 2 equiv. Ag$_2$CO$_3$ in DCM, the desired product **3a** was obtained in 40% yield, along with trace of undesired β-lactam **3aa** generated by the competitive intramolecular C–N bond reductive elimination (entry 1). Toluene was found to be the ideal solvent for this transformation (entry 7, 65% yield). Further screening of additives then established that the addition of I$_2$ could significantly improved the efficiency and **3a** was obtained in

**Table 1 | Screening of reaction conditions.**

| Entry | Solvent | Additive | 3a (%)* | 3aa (%)* |
|---|---|---|---|---|
| 1 | DCE | – | 40 | <5 |
| 2 | MeCN | – | N.D. | N.D. |
| 3 | t-AmOH | – | <5 | N.D. |
| 4 | THF | – | 62 | <5 |
| 5 | MeOH | – | n.d. | <5 |
| 6 | toluene | – | 65 | <5 |
| 7 | toluene | Oxone | 78 | <5 |
| 8 | toluene | BQ | 77 | <5 |
| 9 | toluene | DDQ | <5 | N.D. |
| 10 | toluene | NIS | 18 | <5 |
| **11** | **toluene** | **I$_2$** | **84(76)**† | **<5** |
| 12‡ | toluene | I$_2$ | <5 | 30 |

N.D., not detected.
*Yields were determined by $^1$H NMR using CH$_2$Br$_2$ as the internal standard.
†Isolated yield in parenthesis.
‡1.0 equiv. Ag$_2$CO$_3$ was used.

76% isolated yield (entry 11). It is worth noting that the alkoxycarbonylation reaction was quite sensitive to the amount of Ag$_2$CO$_3$. Attempts to lower the Ag$_2$CO$_3$ loading led to the inhibition of the desired reaction and the competitive intramolecular C–N bond reductive elimination occured predominantly (entry 12, **3a**, <5%; **3aa**, 30%).

**Figure 3 | Pd-catalyzed alkoxycarbonylation of β-methylene C($sp^3$)–H bonds and γ-methyl C($sp^3$)–H bonds.** Reaction conditions: **1** (0.15 mmol), Pd(OAc)$_2$ (10 mol%), Ag$_2$CO$_3$ (2.0 equiv.), I$_2$ (1.0 equiv.), ClCO$_2$R (3.0 equiv.), toluene (2.0 ml), 120 °C, air, 16 h. Isolated yield. [a]20 mol% succinc anhydride was used. [b]140 °C.

**Substrate scope of methylene C($sp^3$)–H alkoxycarbonyaltion.** With the optimal reaction conditions in hand, the scope of this alkoxycarbonylation reaction was investigated (Fig. 3). The reaction was found to be compatible with a broad range of phenylalanine derivatives with various electron-donating and electron-withdrawing substituents (**3b–3q**). Various functional groups, such as methoxy (**3e–3g**), trifluoromethyl (**3h**), acetyl (**3i**), methoxycarbonyl (**3j**), fluoro (**3k** and **3l**), chloro (**3m**) and bromo (**3n**) were tolerated, furnishing the desired products in moderate to good yields. The tolerance of halides was particularly noteworthy since such substituents could serve as versatile handles for further elaboration via cross-coupling. It is noteworthy that arylalanine derivatives (**1b–1q**) were prepared by arylation of the alanine derivative (**5a**) using our previsouly established conditions[40]. Therefore, this protocol also showcases the synthesis of chiral aspartic acid derivatives via a two-step C–H functionalization sequence. Importantly, the alkoxycarbonylation of aliphatic secondary C($sp^3$)–H bonds could also be achieved when 0.2 equiv. of succinic anhydride was included (**3r-3w**). Alkoxycarbonylation of sterically hindered L-leucine derivative containing adjacent secondary alkyl group occurred smoothly under a slightly higher temperature (**3t**, 50%). The

alkoxycarbonylation reaction was found to be highly diastereo-selective, furnishing a single diastereoisomoer as the sole product. The relative and absolute stereochemistry of **3a**, **3d**, **3k** and **4b**, was unambiguously determined by X-ray diffraction, and all other alkoxycarbonylation products were assigned analogously. The *trans* orientation of the *N*-phthaloyl group and the newly incorporated alkoxycarbonyl group was consistent with the proposed stereochemical model and previous reports[40–45,51,52]. In addition, the more remote γ-methyl C($sp^3$)–H bond was also reactive, provided that no reactive β–C–H bonds were present. The alkoxycarbonylation of L-*tert*-Leucine (**3u**), L-isoleucine (**3v**) and L-vlaine (**3w**) proceeded effectively, albeit affording the products in reduced yields.

The scope of the alkyl chloroformates coupling partners was examined subsequently (Fig. 4). The reaction was found to be compatible with a variety of simple and more complex chloroformates. Methoxycarbonylation of phenylanine derivative proceeded smoothly to provide product **4b** in 72% yield. A number of linear alkyl chloroformates gave the corresponding products in good yields (**4b–4e**). Interestingly, the more sterically hindered branched-alkyl chloroformates, such as *i*Pr (**2f**), *i*Bu (**2g**), and cyclopentyl (**2h**) were more reactive, giving the

**Figure 4 | Scope of alkyl chloroformates.** Reaction conditions: **1a** (0.15 mmol), Pd(OAc)$_2$ (10 mol%), Ag$_2$CO$_3$ (2.0 equiv.), I$_2$ (1.0 equiv.), ClCO$_2$R (3.0 equiv.), toluene (2.0 ml), 120 °C, air, 16 h. Isolated yield.

corresponding products in higher yields (**4f–4h**, 82–96% yield). The synthetic potential of this alkoxycarbonylation strategy was further demonstrated by the effective reaction with more complex chloroformates, such as menthyl chloroformate (**2i**) and Fmoc-Cl (**2j**).

**Substrate scope of methyl C($sp^3$)–H alkoxycarbonyaltion.** Next, we sought to investigate whether the alkoxycarbonylation protocol amendable to methyl C($sp^3$)–H bonds. Gratifyingly, the alkoxycarbonylation occurred smoothly to a variety of aliphatic carboxamides bearing $\beta$-methyl C–H bonds with a slightly modified conditions: 10 mol% Pd(TFA)$_2$, 2.0 equiv. Ag$_2$CO$_3$, 1.0 equiv. Na$_3$PO$_4$ and 3.0 equiv. ClCO$_2$Me in toluene at 120 °C. As shown in Fig. 5, N-phthaloyl alanine derivative **5a** reacted effeiently with ClCO$_2$Me in the absence of sodium phosphate, affording the orthogonally protected aspartic acid **6a** in 71% yield. Aliphatic carboxamides bearing either linear chains or cyclohexyl were also compatible with these new conditions (**6b–6f**). A wide range of functional groups, such as ester (**6g**), ethers (Bn-, **6h** and **6l**; Et-, **6k**), alkene (**6i**) and alkyne (**6j**), could also be readily alkoxycarbonylated. The reactions were also tolerant of a number of aryl groups at the $\alpha$, $\beta$- and $\delta$-positions of the carboxamides (**6m–6o** and **6e**). It should be noted that the broad functional group tolerance highlights the synthetic potential of this protocol in late-stage modification and total synthesis of complex molecules[60].

**Synthetic potential.** To further demonstrate the synthetic potential of this reaction, the reaction was conducted in gram scale (Fig. 6). We were pleased to find that the treatment of 3.0 mmol **1a** with ethyl chloroformate gave the corresponding ethoxycarbonylation product **3a** in 74% isolated yield (1.10 g). The competitive intramolecular C–N bond reductive elimination

product **3aa** was also produced in 12% yield when the reaction was scaled up. The desired product **3a** was obtained without any racemization. Moreover, we also found that no epimerization of **3a** has been observed upon prolonged heating under the reaction conditions (Supplementary Figs 8–10).

The ability to easily remove the directing group from the final product is crucial for synthetic applications of this reaction. Previously, we reported that 2-pyridinylisopropyl bidentate auxiliary introduced by us could be removed through a nitrosylation/hydrolysis sequence with a mixture of NaNO$_2$/AcOH/Ac$_2$O (refs 39,40). Baudoin and co-workers has improved the procedure by using NOBF$_4$ as a nitrosation agent and pyridine at low temperature[61,62]. We envisioned that novel process could also be applied to the removal of AQ. As expected, the corresponding carboxylic acid **7** was obtained in 62% yield without further optimization. Following esterification, the corresponding methyl ester **8** was obtained in 91% yield with the retention of configuration (Scheme 3, see Supplementary Methods and Supplementary Fig. 61 for details). It is worth noting that all of the alkoxycarbonylation reactions were operationally simple, without the need for an inert-atmosphere or rigorously moisture-free conditions.

**Mechanistic investigations.** To shed light on the mechanism, several experiments were performed (Fig. 7). First, kinetic isotopic effect (KIE) experiments were conducted by treatment of compound **1a** and its deuterated analogue **1a**-$d_2$ under the standard reaction conditions for 10 min. A $k_H/k_D$ value of 1.5 was obtained in a competitive reaction and 1.7 in parallel reactions on the basis of $^1$H nuclear magnetic resonance (NMR) analysis (Fig. 7a), which is indicative of a secondary kinetic isotope effect. This result also suggests that the cleavage of C–H is not the rate-determining step of the reaction.

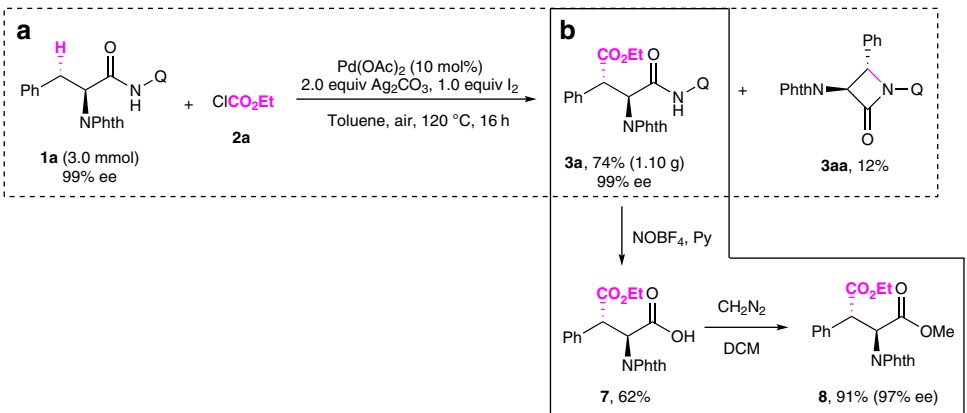

**Figure 5 | Pd-catalysed alkoxycarbonylation of β-Methylene C($sp^3$)–H Bonds.** Reaction conditions: **5** (0.15 mmol), Pd(TFA)$_2$ (10 mol%), Ag$_2$CO$_3$ (2.0 equiv.), Na$_3$PO$_4$ (1.0 equiv.), ClCO$_2$Me (3.0 equiv.), toluene (1.5 ml), 120 °C, air, 20 h. Isolated yield. $^a$In the absence of Na$_3$PO$_4$.

**Figure 6 | Gram-scale synthesis and the removal of directing group.** (**a**) Alkoxycarbonylation of N-phthaloyl phenylalanine derivative **1a** in gram scale. (**b**) The removal of AQ under mild conditions with the retention of configuration.

Second, a stoichiometric reaction of complex **I** with 3 equivalents of ClCO$_2$Et (**2a**) under the optimized reaction conditions was performed, and the alkoxycarbonylated product **3a** was obtained in 40% yield. However, β-lactam **3aa** was produced in 48% yield and no desired product **3a** was observed when complex **I** was treated with iodine in the absence of silver

carbonate (Fig. 7b). These results clearly indicated that the addition of silver carbonate was crucial for the success of this transformation. Although the exact role of the silver salt may simply be a halide scavenger[63–65], it is also proposed to form a bimetallic complex with palladium, which might be important for the reaction[24,27,66].

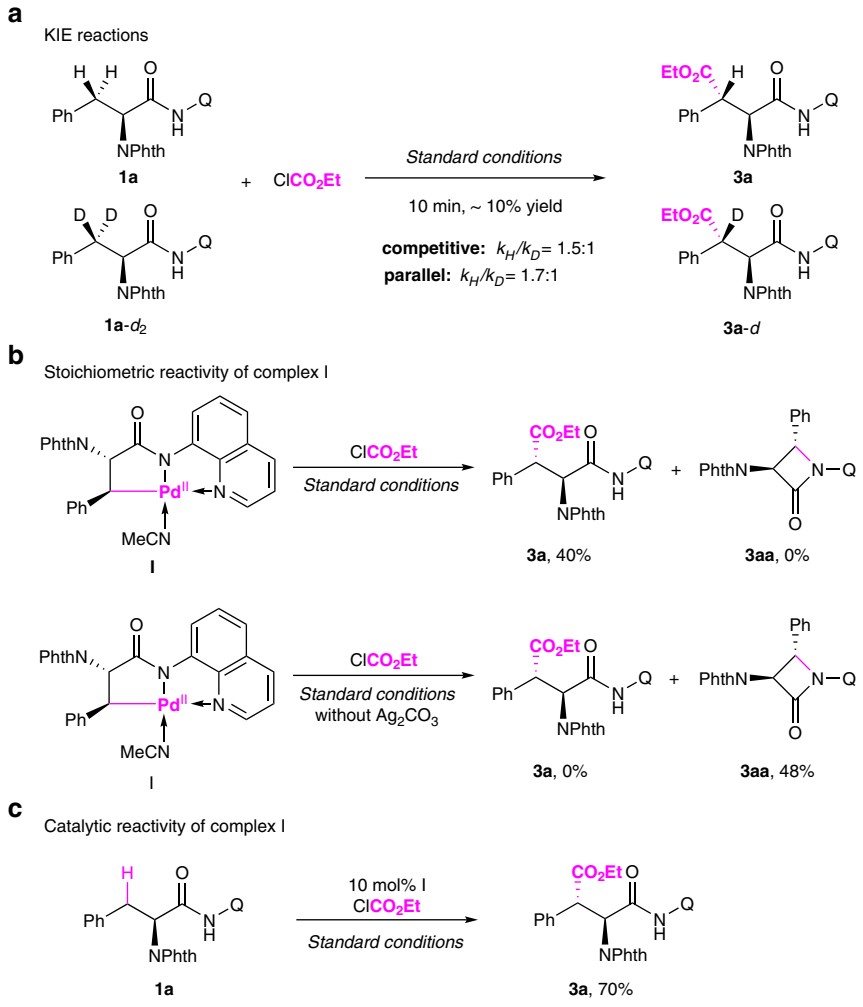

**Figure 7 | Mechanistic Studies.** (**a**) KIE reactions. (**b**) Stoichiometric reactivity of complex **I**. (**c**) Catalytic reactivity of complex **I**.

Finally, we found that palladacycle **I** was a viable precatalyst for the alkoxycarbonylation of **1a**, providing the desired product in 70% yield, which was comparable with the result under the standard conditions (Fig. 7c).

## Discussion

In conclusion, we have developed a new protocol for the direct alkoxycarbonylation of both methylene and methyl C($sp^3$)–H bonds through a Pd(II)/Pd(IV) catalytic cycle. A variety of operationally simple and readily available alkyl chloroformates were used as carbonyl sources. The reaction proceeded with high functional compatibility. Furthermore, this efficient and stereoselective protocol to access orthogonally protected chiral aspartic acid derivatives may find applications in the synthesis of complex molecules. Compared with the well-established C–H carbonylation with CO via Pd(II)/Pd(0) catalysis, the direct alkoxycarbonylation of unactivated C–H bonds through a Pd(II)/Pd(IV) catalytic cycle provides a new mode of reaction and might offer a distinct platform for reaction development. Further studies toward the application of this new strategy to other reaction systems are currently underway.

## Methods

**General methods.** For NMR spectra, high-performance liquid chromatography (HPLC) data, and X-ray analysis of compounds in this manuscript and detailed experimental procedures, see Supplementary Figs 1–67, Supplementary Tables 1–12 and Supplementary Methods. See Supplementary Datasets 1–6 for X-ray CIF files of compounds **III**, **3a**, **3d**, **3k**, **4b** and **6a** (CCDC 1446624, 1487147, 1486639, 1486599, 1446623, 1475241).

**General procedure for secondary C–H alkoxycarbonylation.** To a 50 ml Schlenk tube, were added **1** (0.15 mmol), Pd(OAc)$_2$ (3.5 mg, 0.015 mmol), Ag$_2$CO$_3$ (82.7 mg, 0.3 mmol), I$_2$ (38.0 mg, 1.0 equiv.), ClCO$_2$R (0.45 mmol, 3.0 equiv.) and toluene (2.0 ml). The tube was sealed under air. The mixture was stirred at room temperature for 5 min then heated at 120 °C for 16 h. After cooling to room temperature, the reaction mixture was diluted with EtOAc (10 ml) and filtered through a pad of Celite. After concentration in vacuo, the crude reaction mixture was purified by silica gel flash chromatography.

**General Procedure for Primary C–H Alkoxycarbonylation.** To a 50 ml Schlenk tube, were added **1** (0.15 mmol), Pd(OTFA)$_2$ (5.0 mg, 0.015 mmol), Ag$_2$CO$_3$ (82.7 mg, 0.3 mmol), Na$_3$PO$_4$ (49.0 mg, 0.3 mmol), ClCO$_2$Me (0.45 mmol, 3.0 equiv.) and toluene (2.0 ml). The tube was sealed under air. The mixture was stirred at room temperature for 5 min then heated at 120 °C for 20 h. After cooling to room temperature, the reaction mixture was diluted with EtOAc (10 ml) and filtered through a pad of Celite. After concentration in vacuo, the crude reaction mixture was purified by silica gel flash chromatography.

**Data availability.** The X-ray crystallographic structures for compounds **III**, **3a**, **3d**, **3k**, **4b**, **6a** reported in this article have been deposited at the Cambridge Crystallographic Data Centre (CCDC), with the accession codes CCDC 1446624, 1487147, 1486639, 1486599, 1446623, 1475241 (http://www.ccdc.cam.ac.uk/data_request/cif). The authors declare that all other relevant data supporting the findings of this study are available within the article and its Supplementary Information files.

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

## Acknowledgements

Financial support from the National Basic Research Program of China (2015CB856600) and the NSFC (21572201, 21422206, 21272206) is gratefully acknowledged. We thank Prof. Xueqian Kong from Zhejiang University, China for help with the use of facilities in his lab.

## Author contributions

G.L. and B.-F.S. conceived and designed the study. G.L. principally performed the experiments. X.-S.Y., K.C., Q.Z. and S.-Q.Z. helped to conduct some experiments and collect data. B-F.S. provided overall supervision and wrote the manuscript.

## Additional information

**Competing financial interests:** B.-F.S., G.L. and X.-S.Y. are the co-inventors on a patent application: 'A highly efficient method to the synthesis of aspartate derivatives' CN application number 201610147639.1. The remaining authors declare no competing financial interests.

**How to cite this article**: Liao, G. *et al.* Stereoselective alkoxycarbonylation of unactivated C(*sp*$^3$)–H bonds with alkyl chloroformates via Pd(II)/Pd(IV) catalysis. *Nat. Commun.* 7:12901 doi: 10.1038/ncomms12901 (2016).

