## [Peer review file · Nature Communications]

Reviewers' comments:

Reviewer #1 (Remarks to the Author):

Shi and co-workers disclosed a Pd-catalyzed protocol for alkoxyacylation of C(sp³)-H bonds with chloroformates. To ensure regioselectivity of the functionalization, the authors employed 8-aminoquinoline as the auxiliary for cyclopalladacycle formation. The apparent preference of a five-membered ring formation resulted in high degree of regiocontrol being achieved here. Indeed, the auxiliary and its related chemistry have been demonstrated by Daugulis and co-workers some years ago. This work was initiated by a proof-of-concept study involving a stoichiometric reaction of a well-defined cyclopalladacyclic complex derived from phenylalanine and chloroformate; the desired products were obtained in modest yields (ca. 20%). Yet, the authors proceeded to reaction optimization and significant results were obtained. Notably, the best results were achieved with the use of two equivalents of silver carbonate and one equivalent of iodine. Yet, other oxidants such as Oxone and BQ were also found to give comparable yields in the optimization study.

In the substrate scope study, the authors demonstrated some salient features of this Pd-catalyzed alkoxyacylation protocol. Firstly, some useful polar functional groups (e.g. Br, acetyl, methoxy, esters) on the aryl rings were tolerated. Secondary C(sp³)-H bonds can also be functionalized as shown in the leucine example. Interestingly, the successful functionalization of the secondary C(sp³)-H bond requires the use of 20 mol% of succinic anhydride. More remote gamma methyl C-H bond has also been functionalized albeit with lower yield. Chloroformates other than the methyl ester were also coupled successfully to phenylalanine. As expected, the methyl C(sp³)-H bonds of a series of aliphatic carboxylic acids can be functionalized.

The Pd-catalyzed reaction has been tested on a gram-scale, and the desired product was obtained in reasonable yield, implying a potential application of this reaction for preparation-scale synthesis. Easy removal of the auxiliary using a new method (NONF4) is attractive for preserving other protecting groups.

The substrate scope study was followed by a brief mechanistic investigation. The alkoxyacylation does not exhibit significant primary H/D kinetic isotope effect. It was concluded that C-H bond cleavage should not be turnover limiting. Stoichiometric reaction of the cyclopalladated complex of phenylalanine with chloroformate confirmed the involvement of the alkylpalladium(II) complexes. Yet, further details on the coupling reaction were not addressed in this work.

In general, this work is a novel discovery of a potentially important C-H functionalization reaction. In the Introduction section, a Pd(II)/Pd(IV) reaction manifold was expected to operate. The authors seems to follow the earlier chemistry developed for the same system (i.e., cyclopalladacycle with 8-aminoquinoline as auxiliary) in developing the current alkoxyacylation. The use of chloroformate seems to be a logical electrophilic reagent for Pd(IV) generation. Yet, any evidence for Pd(IV)-acyl complex is lacking in this work. The rather thin mechanistic study, in my opinion, should not weaken the significance of this

work. The manuscript is well organized and easy to follow. I would recommend publication of this work after the authors addressing the following minor points:

(1) To begin, the authors studied the carbonylation of the cyclopalladated complex of phenylalanine. No carbonylation to the Pd-C bond was observed. Yet, the group of JQ Yu reported some years ago the carbonylation of aliphatic C-H bonds to form lactam in JACS. In this work, perhaps the three-coordinated Pd(II) center is not reactive for CO migratory insertion. The Pd(IV) center, if generated, should be active for migratory insertion, as suggested by Yu's finding. Assuming Pd(IV) is involved in the current alkoxy carbonylation reaction, did the authors observe any de-carbonylation products? How important of the de-carbonylation reaction, if indeed observed?

(2) For those lower-yielding examples (e.g. 3u, 3v, 3w, 6n, 6g, 6o..), are there any side-products formed? What are these side-products, if any?

Reviewer #2 (Remarks to the Author):

This manuscript by Bing-Feng Shi and co-workers describes the palladium(II)-catalyzed alkoxy carbonylation of unactivated primary and secondary C(sp³)-H bonds. To achieve this transformation, they employed the popular aminoquinoline directing group, initially developed by Daugulis and co-workers, which has proven successful in numerous catalytic C-H functionalization reactions. As pointed out by the authors in the introduction, the carbonylation of primary C-H bonds has been reported by several groups (refs #21-26), but its extension to secondary C-H bonds is more challenging and remains uncharted. At the onset of this work, the authors show that the isolated palladacycle, obtained from a stoichiometric C-H activation reaction, fails to undergo carbonyl insertion in the case of secondary C-H bonds (Figure 2). This important observation led to the development of the current method employing chloroformates as CO surrogates, which allow to access C-H acylation products. Reaction conditions were optimized for the acylation of both secondary (Table 1, Figures 3-4) and primary (Figure 5) C-H bonds. The scope is particularly impressive, and in the case of secondary C-H bonds the anti diastereoisomer is formed exclusively, as can be expected from the proposed mechanism and from related C-H functionalization reactions. The formed ester products are very useful building amino acid blocks which may find use in the synthesis of peptides and other bioactive molecules. The quality of the experimental part is very good. In conclusion, I recommend acceptance in Nature Communications with the minor revision noted below.

1. The term "retention of chirality", which is employed several times, should be replaced with "retention of configuration".
2. On several occasions, the authors make a strong case for chloroformate reagents instead of the "toxic, explosive, and gaseous carbon monoxide". The use of CO may be a problem for small scale laboratory applications, but is not a big issue at the industrial scale (see the oxo process!) where it is economically advantageous. Please moderate this argument.
3. Please comment on the possible role of iodine as additive in the acylation of secondary C-H bonds. What happens exactly when the isolated palladacycle I is treated with iodine

without Ag₂CO₃ (Figure 7b)? Product 3a is not formed, but is the palladacycle intact? Is beta-lactam 3aa also observed?

4. Substrate scope (Figure 3):

- it would be worth to mention that the arylalanine reactants (1a-q)) were prepared by directed C-H arylation of an alanine derivative; ultimately this is a nice two-step C-H functionalization sequence;

- I assume that the anti configuration of the acylation products was ascribed by analogy with 4b, for which an X-ray structure is described (Figure 2). This should be mentioned. An additional proof of configuration would be useful for non-aromatic products (3r-w);

- the products, especially 3a-q, are potentially epimerizable under basic conditions, and I find it quite remarkable that a single diastereoisomer was obtained. Has any epimerization been observed upon prolonged heating under the reaction conditions? Please comment.

5. On top of p. 12 it is stated that "These results demonstrated the secondary kinetic isotope effect...". I would be more cautious with the interpretation of these results, and write "A k_H/k_D value of 1.5...on the basis of ¹H NMR (Figure 7a), which is indicative of a secondary kinetic isotope effect. This result also suggests that the cleavage of the C-H bond is not the rate-determining step of the reaction".

6. There are a number of typos and grammatical errors and they should be corrected before publication.

Reviewer #3 (Remarks to the Author):

Shi describes the chelation-assisted palladium-catalyzed C-H alkoxyacylation. The bidentate quinolineamide directing group was used for the functionalization of C(sp³)-H bonds. Chloroformates enabled overall alkoxyacylations as an alternative to the use of cheap CO. The optimized catalyst was broadly applicable and mechanistic studies suggested a Pd(II)/Pd(IV) catalytic cycle.

Chloroformates were already used in metal-catalyzed C-H activation chemistry, for example by Kakiuchi. Various palladium-catalyzed C(sp³)-H functionalization were achieved with the quinolineamide directing group, most notably by Daugulis. This also holds true for palladium-catalyzed C-H arylations and alkylations.

Overall, I feel that the new manuscript is rather an extension of the previous reports, and lacks elements of novelty that warrant publication in Nature Communications.

June 28, 2016

Manuscript ID: NCOMMS-16-09282

Title: "Stereoselective Alkoxycarbonylation of Unactivated C(sp³)-H Bonds with Alkyl Chloroformates via Pd(II)/Pd(IV) Catalysis"

Author(s): Gang Liao, Xue-Song Yin, Kai Chen, Qi Zhang, Shuo-Qing Zhang, Bing-Feng Shi

Dear Respected Reviewers:

Thank you for the comments regarding our manuscript entitled "Stereoselective Alkoxycarbonylation of Unactivated C(sp³)-H Bonds with Alkyl Chloroformates via Pd(II)/Pd(IV) Catalysis". All the suggestions were highly encouraging and constructive. We sincerely thank you for your time and kind suggestions, and with your help, we could now significantly improve our manuscript. We are now submitting the revised manuscript after fully addressing those comments as shown below.

For your convenience, a detailed point-by-point list of specific changes and comments is provided below:

Reviewer 1

This reviewer recommended the publication of this work after the authors addressing the following minor points:

(1) To begin, the authors studied the carbonylation of the cyclopalladated complex of phenylalanine. No carbonylation to the Pd-C bond was observed. Yet, the group of JQ Yu reported some years ago the carbonylation of aliphatic C-H bonds to form lactam in JACS. In this work, perhaps the three-coordinated Pd(II) center is not reactive for CO migratory insertion. The Pd(IV) center, if generated, should be active for migratory insertion, as suggested by Yu's finding. Assuming Pd(IV) is involved in the current alkoxycarbonylation reaction, did the authors observe any de-carbonylation products? How important of the de-

carbonylation reaction, if indeed observed?

Thank you for the comments and the instructive question. We haven't observed any de-carbonylation products under the standard reaction conditions. We have also tried to prolong the reaction time to 48 hours by the use of phenylalanine derivative **1a** as a model substrate. As expected, no de-carbonylation product was detected.

(2) For those lower-yielding examples (e.g. 3u, 3v, 3w, 6n, 6g, 6o..), are there any side-products formed? What are these side-products, if any?

Thank you for the comments. We didn't observe any side-products and the starting materials were recovered accordingly (**1u**, 31%; **1v**, 29%; **1w**, 34%; **5n**, 41%; **5g**, 49%; **5o**, 48%).

Reviewer 2

This reviewer recommended acceptance in Nature Communications with the minor revision.

1. The term "retention of chirality", which is employed several times, should be replaced with "retention of configuration".

Thank you for the suggestions. We have replaced the term "retention of chirality" with "retention of configuration". Please refer to the revised manuscript.

2. On several occasions, the authors make a strong case for chloroformate reagents instead of the "toxic, explosive, and gaseous carbon monoxide". The use of CO may be a problem for small scale laboratory applications, but is not a big issue at the industrial scale (see the oxo process!) where it is economically advantageous. Please moderate this argument.

Thank you for the suggestion and we agree with the reviewer that the use of carbon monoxide is only a problem for small laboratory-scale reactions. We have moderate it in the revised manuscript (Page 2, second paragraph): "...the use of CO is still relatively inconvenient on

laboratory-scale due to its gaseous form, toxic nature and flammability.” In addition, we have also moderate the assertions in graphic abstract and Figure 1a as “CO: toxic, gaseous, inconvenient on laboratory-scale”. Please refer to the revised manuscript for details.

3. Please comment on the possible role of iodine as additive in the acylation of secondary C-H bonds. What happens exactly when the isolated palladacycle I is treated with iodine without Ag₂CO₃ (Figure 7b)? Product 3a is not formed, but is the palladacycle intact? Is beta-lactam 3aa also observed?

Thank you for the comments. Although the exact role of iodine is unclear at this point, we rationalize that iodine may act both as an electron donor ligand to the Pd-center and as a co-oxidant. We did not observe the desired product **3a** when palladacycle **I** was treated with iodine without Ag₂CO₃ (Figure 7b), the β -lactam **3aa**, however, was formed in 48% yield. We have added one sentence in Page 12: “However, β -lactam **3aa** was produced in 48% yield and no desired product **3a** was observed when complex **I** was treated with iodine in the absence of silver carbonate (Figure 7b).”.

Figure 7b.

4. Substrate scope (Figure 3):

- it would be worth to mention that the arylalanine reactants (**1a-q**) were prepared by directed C-H arylation of an alanine derivative; ultimately this is a nice two-step C-H functionalization sequence;

Thank you for the suggestions. We have mentioned the two-step C-H functionalization sequence of alanine derivative in the revised manuscript (Page 7, paragraph 1): “It is noteworthy that arylalanine derivatives (**1b-1q**) were prepared by arylation of the alanine derivative (**5a**) using our

previously established conditions.³⁹ Therefore, this protocol also showcases the synthesis of chiral aspartic acid derivatives via a two-step C–H functionalization sequence.”.

- I assume that the anti configuration of the acylation products was ascribed by analogy with 4b, for which an X-ray structure is described (Figure 2). This should be mentioned. An additional proof of configuration would be useful for non-aromatic products (3r-w);

Thank you for the suggestions. We tried to get more single-crystals to confirm the characterization. Gratifyingly, single-crystals of compounds **3a**, **3d**, and **3k** suitable for X-ray diffraction analysis have been obtained, which unambiguously confirmed that the *N*-phthaloyl group and the newly incorporated alkoxy carbonyl group are oriented *anti* to one another. Please refer to the revised manuscript (Figure 2e and Figure 3) for details. Unfortunately, we can't get any single-crystals for the non-aromatic products (**3r-t**), since these compounds are oil. We have added one sentence in Page 7: “The relative and absolute stereochemistry of **3a**, **3d**, **3k**, and **4b**, was unambiguously determined by X-ray diffraction, and all other alkoxy carbonylation products were assigned analogically. The *trans* orientation of the *N*-phthaloyl group and the newly incorporated alkoxy carbonyl group was consistent with the proposed stereochemical model and previous reports.^{39-44,50,51}”. CCDC numbers for compounds **3a**, **3d** and **3k** have been updated in Additional Information (page 21): “**Accession codes:** The X-ray crystallographic structures for compounds **III**, **3a**, **3d**, **3k**, **4b**, **6a** reported in this article have been deposited at the Cambridge Crystallographic Data Centre (CCDC), under deposition number CCDC 1446624, 1487147, 1486639, 1486599, 1446623, 1475241. These data can be obtained free of charge from the Cambridge Crystallographic Data Centre via http://www.ccdc.cam.ac.uk/data_request/cif.” Please also refer to the revised SI (Supplementary Figs. 63-65, Supplementary Tables 8-10) for the data.

Figure 2e: Stoichiometric reaction of palladacycle **I** with ClCO₂R. The structure of compounds **3a** and **4b** was confirmed by single crystal X-ray diffraction.

Figure 3

Supporting Information Page S62-S63:

Supplementary Figure 63. Xay for complex 3a

Supplementary Figure 64. Xay for complex 3d

Supplementary Figure 65. Xay for complex 3k

Supporting Information Page S71-S73:

Supplementary Table 8 Crystal data and structure refinement for 3a

	Bond precision:	C-C = 0.0051 Å	Wavelength= 0.71073
Cell:	$a = 7.7318 (4)$	$b = 14.2451 (8)$	$c = 21.9203 (17)$
	$\alpha = 90$	$\beta = 90$	$\gamma = 90$
Temperature:	171 K		
	Calculated		Reported
Volume	2414.3 (3)		2414.3 (3)
Space group	P 21 21 21		P 21 21 21
Hall group	P 2ac 2ab		P 2ac 2ab

Moiety formula	C29 H23 N3 O5	C29 H23 N3 O5
Sum formula	C29 H23 N3 O5	C29 H23 N3 O5
Mr	493.50	493.50
Dx, g cm ⁻³	1.358	1.358
Z	4	4
Mu (mm ⁻¹)	0.094	0.094
F000	1032.0	1032.0
F000'	1032.49	
h, k, l _{max}	9, 17, 26	9, 17, 26
N _{ref}	4440 [2546]	2540
T _{min} , T _{max}	0.956, 0.979	0.891, 1.000
T _{min} '	0.955	
Correction method= # Reported T Limits: T _{min} =0.891 T _{max} =1.000		
AbsCorr = MULTI-SCAN		
Data completeness= 1.00 /0.57		Theta(max)= 25.350
R(reflections)= 0.0414(2106)		wR2(reflections)= 0.1079(2540)
S = 1.124		N _{par} = 335

Supplementary Table 9 Crystal data and structure refinement for 3d

Bond precision:	C-C = 0.0072 Å	Wavelength=0.71073
Cell:	a = 9.2583 (12) b = 14.0257 (14) c = 12.1921 (12)	
	α = 90 β = 110.498 (12) γ = 90	
Temperature:	293 K	
	Calculated	Reported
Volume	1483.0 (3)	1483.0 (3)
Space group	P 21	P 1 21 1
Hall group	P 2yb	P 2yb
Moiety formula	C33 H31 N3 O5	C33 H31 N3 O5
Sum formula	C33 H31 N3 O5	C33 H31 N3 O5
Mr	549.61	549.61
Dx, g cm ⁻³	1.231	1.231
Z	2	2
Mu (mm ⁻¹)	0.084	0.084
F000	580.0	580.0
F000'	580.27	
h, k, l _{max}	11, 16, 14	11, 16, 14
N _{ref}	5423 [2831]	2816
T _{min} , T _{max}	0.960, 0.973	0.963, 1.000
T _{min} '	0.960	
Correction method= # Reported T Limits: T _{min} =0.963 T _{max} =1.000		
AbsCorr = MULTI-SCAN		
Data completeness= 0.99/0.52		Theta(max)= 25.350
R(reflections)= 0.0509 (2207)		wR2(reflections)= 0.1250(2816)
S = 1.091		N _{par} = 406

Supplementary Table 10 Crystal data and structure refinement for 3k

Bond precision:	C-C = 0.0044 Å	Wavelength=0.71073
Cell:	a = 7.7205 (4) b = 14.3320 (9) c = 22.282 (1)	
	α = 90 β = 90 γ = 90	
Temperature:	293 K	

	Calculated	Reported
Volume	2465.5 (2)	2465.5 (2)
Space group	P 21 21 21	P 21 21 21
Hall group	P 2ac 2ab	P 2ac 2ab
Moiety formula	C29 H22 F N3 O5	C29 H22 F N3 O5
Sum formula	C29 H22 F N3 O5	C29 H22 F N3 O5
Mr	511.50	511.50
Dx, g cm ⁻³	1.378	1.378
Z	4	4
Mu (mm ⁻¹)	0.101	0.101
F000	1064.0	1064.0
F000'	1064.56	
h, k, l _{max}	9, 17, 26	9, 17, 26
N _{ref}	4524 [2592]	2585
T _{min} , T _{max}	0.952, 0.960	0.922, 1.000
T _{min} '	0.952	
Correction method= # Reported T Limits: T _{min} =0.922 T _{max} =1.000		
AbsCorr = MULTI-SCAN		
Data completeness= 1.00/0.57		Theta(max)= 25.350
R(reflections)= 0.0369 (2103)		wR2(reflections)= 0.0983(2585)
S = 1.057		N _{par} = 344

- the products, especially 3a-q, are potentially epimerizable under basic conditions, and I find it quite remarkable that a single diastereoisomer was obtained. Has any epimerization been observed upon prolonged heating under the reaction conditions? Please comment.

Thank you for the suggestions, we have carried out the reaction of **1a** under the reaction conditions for 16, 24, and 48 h respectively, and no epimerization of **3a** has been observed. Please refer to the revised manuscript (Page 11): “Moreover, we also found that no epimerization of **3a** has been observed upon prolonged heating under the reaction conditions (see Supplementary Figs. 8-10).” and the revised supporting information for details (Supplementary Figs. 8-10, Page S8-S10; Supplementary Table 6, Page S69; Page S84-S85).

Supporting Information Page S8-S10:

AD-H, n-hex : iPrOH/55/45, 0.9 ml/min, 220 nm
 分析者 : System Administrator
 样品名 : L-16
 样品ID :
 进样体积 : 10
 数据文件 : L-16_1cd
 方法文件 : 13_1cm
 报告格式文件 : st_1sr
 分析日期/时间 : 2016-6-16 16:23:22
 处理日期/时间 : 2016-6-16 17:04:08
 重复进样计数 : 1
 备注 : AD-H, n-hex : iPrOH/55/45, 0.9 ml/min, 220 nm

峰表

峰号	保留时间	面积	高度	标记	面积%
1	20.025	82612193	2002353	H	99.416
2	35.668	17437	4888		0.254
总计		90266210	2009241		100.000

Supplementary Figure 8. HPLC spectrum for 3a (reaction time: 16 h)

AD-H, n-hex : iPrOH/55/45, 0.9 ml/min, 220 nm
 分析者 : System Administrator
 样品名 : LGS-44-1
 样品ID :
 进样体积 : 10
 数据文件 : LGS-44-1_1cd
 方法文件 : 13_1cm
 报告格式文件 : st_1sr
 分析日期/时间 : 2016-6-16 17:05:07
 处理日期/时间 : 2016-6-16 17:51:15
 重复进样计数 : 1
 备注 : AD-H, n-hex : iPrOH/55/45, 0.9 ml/min, 220 nm

峰表

峰号	保留时间	面积	高度	标记	面积%
1	20.337	97212973	1951351		99.511
2	35.337	616151	1996		0.656
总计		96929124	1953347		100.000

Supplementary Figure 9. HPLC spectrum for 3a (reaction time: 24 h)

AD-H, n-hex : iPrOH/55/45, 0.9 ml/min, 220 nm
 分析者 : System Administrator
 样品名 : LGS-44-2
 样品ID :
 进样体积 : 5.5
 数据文件 : LGS-44-2_1cd
 方法文件 : 13_1cm
 报告格式文件 : st_1sr
 分析日期/时间 : 2016-6-16 22:27:25
 处理日期/时间 : 2016-6-16 23:08:15
 重复进样计数 : 1
 备注 : AD-H, n-hex : iPrOH/55/45, 0.9 ml/min, 220 nm

峰表

峰号	保留时间	面积	高度	标记	面积%
1	21.133	83669013	1011800		99.268
2	35.782	967088	8924		0.232
总计		84636101	1020724		100.000

Supplementary Figure 10. HPLC spectrum for 3a (reaction time: 48 h)

Supplementary Table 6 Ee Value of 3a with Prolonged Heating

entry	time (h)	yield (%)	ee (%)
1	16	76	99
2	24	51	99
3	48	50	99

Supporting Information Page S84-S85:

(2S,3S)-Ethyl 3-(1,3-dioxoisindolin-2-yl)-4-oxo-2-phenyl-4-(quinolin-8-ylamino)butanoate (3a)

1) Reaction time: 16 h

Phenylalanine derivative **1a** (99% ee) was used as the starting material. The title compound **3a** was prepared under the optimized conditions (16 h) and purified by column chromatography (petroleum ether: dichloromethane: ethyl acetate = 7: 1: 2, R_f = 0.5). **3a** was obtained as a pale yellow solid (56.3 mg, 76%). $[\alpha]_D^{20} = -127.5$ (1.0 M in CHCl_3); $^1\text{H NMR}$ (400 MHz, CDCl_3) δ 10.17 (s, 1H), 8.70 (dd, $J = 6.8, 2.1$ Hz, 1H), 8.52 (dd, $J = 4.2, 1.5$ Hz, 1H), 8.07 (dd, $J = 8.3, 1.5$ Hz, 1H), 7.72 (dd, $J = 5.5, 3.1$ Hz, 2H), 7.63 (dd, $J = 5.5, 3.1$ Hz, 2H), 7.52 – 7.42 (m, 2H), 7.37 – 7.28 (m, 3H), 7.17 (t, $J = 7.3$ Hz, 2H), 7.11 (t, $J = 7.2$ Hz, 1H), 6.04 (d, $J = 11.4$ Hz, 1H), 5.04 (d, $J = 11.5$ Hz, 1H), 4.29 (dq, $J = 10.8, 7.1$ Hz, 1H), 4.18 (dq, $J = 10.8, 7.1$ Hz, 1H), 1.23 (t, $J = 7.1$ Hz, 3H). $^{13}\text{C NMR}$ (101 MHz, CDCl_3) δ 172.0, 167.4, 166.0, 148.4, 138.5, 136.3, 134.5, 134.3, 133.8, 131.3, 128.8, 128.7, 128.1, 127.9, 127.3, 123.6, 122.1, 121.7, 117.0, 61.6, 55.8, 50.0, 14.1. IR (neat): ν 3334, 2924, 2852, 1778, 1722, 1692, 1529, 1487, 1466, 1430 cm^{-1} ; HRMS (ESI): calc. for $\text{C}_{29}\text{H}_{23}\text{N}_3\text{O}_5$ ($\text{M}+\text{H}^+$): 494.1710; Found: 494.1711. HPLC Chiralpak[®] AD-H column, n-hexane/isopropanol = 55:45, flow rate = 0.90 mL/min, $\lambda = 220$ nm, 20.6 (minor), 36.5 (major), 99% ee.

2) Reaction time: 24 h

Phenylalanine derivative **1a** (99% ee) was used as the starting material. The title compound **3a** was prepared under the optimized conditions (24 h) and purified by column chromatography (petroleum ether: dichloromethane: ethyl acetate = 7: 1: 2, R_f = 0.5). **3a** was obtained as a pale yellow solid (37.8 mg, 51%). HPLC Chiralpak[®] AD-H column, n-hexane/isopropanol = 55:45, flow rate = 0.90 mL/min, $\lambda = 220$ nm, 20.4 (minor), 38.4 (major), 99% ee.

3) Reaction time: 48 h

Phenylalanine derivative **1a** (99% ee) was used as the starting material. The title compound **3a** was prepared under the optimized conditions (48 h) and purified by column chromatography (petroleum ether: dichloromethane: ethyl acetate = 7: 1: 2, R_f = 0.5). **3a** was obtained as a pale yellow solid (37.1 mg, 50%). HPLC Chiralpak[®] AD-H column, n-hexane/isopropanol = 55:45, flow rate = 0.90 mL/min, $\lambda = 220$ nm, 21.5 (minor), 38.8 (major), 99% ee.

5. On top of p. 12 it is stated that "These results demonstrated the secondary kinetic isotope effect...". I would be more cautious with the interpretation of these results, and write "A

k_H/k_D value of 1.5...on the basis of 1H NMR (Figure 7a), which is indicative of a secondary kinetic isotope effect. This result also suggests that the cleavage of the C-H bond is not the rate-determining step of the reaction".

Thank you for the suggestions and we agree with the reviewer. We have changed it: “A k_H/k_D value of 1.5 was obtained in a competitive reaction and 1.7 in parallel reactions on the basis of 1H NMR analysis (Figure 7a), which is indicative of a secondary kinetic isotope effect. This result also suggests that the cleavage of C–H is not the rate-determining step of the reaction.”. Please check the revised manuscript for details (Page 12).

6. There are a number of typos and grammatical errors and they should be corrected before publication.

Thank you for the suggestions. We have corrected the typos and grammatical errors. Please check the revised manuscript for details.

Thank you very much for your time and consideration. I'm looking forward to hearing from you.

Sincerely Yours,

Bing-Feng Shi

REVIEWERS' COMMENTS:

Reviewer #2 (Remarks to the Author):

In this revised version, the authors addressed most of my comments and concerns and hence I am happy to propose acceptance in Nature Communications.

Department of Chemistry
ZHEJIANG UNIVERSITY

Prof. Bing-Feng Shi
Department of Chemistry
Zhejiang University, Hangzhou 310027 (China)
Email: bfshi@zju.edu.cn
Tel: +86-571-87951352
Fax: +86-571-87951895

July 26, 2016

Manuscript ID: NCOMMS-16-09282A

Subject: Response to referees

Title: "**Stereoselective Alkoxy carbonylation of Unactivated C(sp³)-H Bonds with Alkyl Chloroformates via Pd(II)/Pd(IV) Catalysis**"

Author(s): Gang Liao, Xue-Song Yin, Kai Chen, Qi Zhang, Shuo-Qing Zhang, Bing-Feng Shi

REVIEWERS' COMMENTS:

Reviewer #2 (Remarks to the Author):

Comment: In this revised version, the authors addressed most of my comments and concerns and hence I am happy to propose acceptance in Nature Communications.

Response: Referee 2 agree that we have addressed his comments and recommend the acceptance in Nature Communications. We thank you all the referees and the editors for their time and consideration.